# Environmental Adaptation and Differential Replication in Machine Learning

**DOI:** 10.3390/e22101122

**Published:** 2020-10-03

**Authors:** Irene Unceta, Jordi Nin, Oriol Pujol

**Affiliations:** 1BBVA Data & Analytics, 28050 Madrid, Spain; irene.unceta@bbvadata.com; 2Department of Mathematics and Computer Science, Universitat de Barcelona, 08007 Barcelona, Spain; oriol_pujol@ub.edu; 3Department of Operations, Innovation and Data Sciences, Universitat Ramon Llull, ESADE, 08172 Sant Cugat del Vallès, Spain

**Keywords:** natural selection, differential replication, machine learning, knowledge distillation, editing, copying

## Abstract

When deployed in the wild, machine learning models are usually confronted with an environment that imposes severe constraints. As this environment evolves, so do these constraints. As a result, the feasible set of solutions for the considered need is prone to change in time. We refer to this problem as that of environmental adaptation. In this paper, we formalize environmental adaptation and discuss how it differs from other problems in the literature. We propose solutions based on differential replication, a technique where the knowledge acquired by the deployed models is reused in specific ways to train more suitable future generations. We discuss different mechanisms to implement differential replications in practice, depending on the considered level of knowledge. Finally, we present seven examples where the problem of environmental adaptation can be solved through differential replication in real-life applications.

## 1. Survival of the Fittest

“If during the long course of ages and under varying conditions of life, organic beings vary at all in the several parts of their organization, [...] I think it would be a most extraordinary fact if no variation ever had occurred useful to each being’s own welfare, in the same way as so many variations have occurred useful to man. However, if variations useful to any organic being do occur, assuredly individuals thus characterized will have the best chance of being preserved in the struggle for life; and from the strong principle of inheritance they will tend to produce offspring similarly characterized. This principle of preservation, I have called, for the sake of brevity, Natural Selection.” [Charles Darwin, Origin of the Species, p. 127, 1859]

Natural selection explores how organisms adapt to a changing environment in their struggle for survival [1]. Survival in this context is intrinsically defined by a complex and generally unknown fitness function that governs the life of all living creatures. The closer they move towards the optimal value of this function, the better fit they are to face the hard conditions imposed by their environment and, hence, the better chance they have at survival.

This predominant role of the environment is not unique to living organisms. It is also present in aspects of human society, from business to culture, including everything from economic changes, adjustment of moral and ethical concerns, regulatory revisions or the reframing of societal rules that results from unexpected global crises or natural catastrophes. In a smaller scale, it also affects machine learning model deployment. Indeed, the success or failure of a predictive model is largely influenced by its immediate surroundings. Not in vain did the Gartner Data Science Team Survey [2] find that over 60% of machine learning models designed and trained in companies during 2018 are never actually served into production, due mostly to a failure to meet the constraints imposed by their immediate environment. Hence, it seems reasonable to assume that understanding this environment is a necessary first step when devising any industrial machine learning solution.

A machine learning model’s environment comprises all the elements that interact with the model throughout its lifespan, including the data and their different sources, the deployment infrastructure, the governance protocol, or the regulatory framework. The elements may be both internal and external to a company. nternal elements refer to those, such as the feature engineering process or the deployment infrastructure, that are controlled by the data scientists and which are related, to a certain extent, to their strategic decisions. External elements, on the other hand, come from outside the company itself and therefore generally out of its control. They refer, for example, to the trends of the market, the behavior of consumers, the relationship with third parties or any other aspect that may affect a machine learning based product or service. Both internal and external components impose requirements on how models are designed, trained and served into production. A machine learning model’s environment can therefore be understood as a set of constraints. Moreover, given that these requirements are prone to change in time, a machine learning model’s environment is generally dynamic, rather than static. This means that the requirements that constrain a model usually evolve throughout its lifespan. To survive in such an environment and ensure a sustained delivery over time, machine learning models need to adapt to new conditions.

This idea of adaptation has been present in the literature since the early times of machine learning, as practitioners have had to devise ways in which to adapt theoretical proposals to their everyday-life scenarios [3,4,5]. As the discipline has evolved, so have the techniques available to this end. Consider, for example, situations where the underlying data distribution changes resulting in a concept drift. While traditional batch learners are incapable of adapting to such drifts, online learning algorithms can iteratively update their knowledge according to changes in the data [6]. Alternatively, transfer learning also studies solutions to changes in the data distribution. In particular, it focuses on cases where learning a given task can be improved through the transfer of knowledge acquired when learning a related task [7,8,9]. In certain transfer learning problems, the change of task is accompanied by a change in domain, so that data labelled in a single [10] or multiple [11] source domains are leveraged to learn a classifier on unseen data in another domain. In all these cases, the original solution needs to be adapted to the new domain or task. Yet, the defined hypothesis space remains feasible. There are situations, however, where it is not the data distributions or the problem domain that change, but a model’s environment itself, rendering the existing solution obsolete or even unfeasible.

Say that one of the original input attributes is no longer available, that a deployed black-box solution is required to be interpretable or that updated software licenses require moving our current machine learning system to a new production environment. These changes generally require the definition of a new model in a different hypothesis space. Consider, for example, cases where there is a change in the business needs. Commercial machine learning applications are designed to answer very specific business objectives that may evolve in time. See, for example, fraud detection algorithms [12], which need to be regularly retrained to incorporate new types of fraud. In doing so, a new feasible set of solutions is required. A straightforward solution in this context is to discard the existing model and re-train another in a new space. A main drawback of this approach, however, is that in discarding the existing solution altogether, we also discard all the knowledge it acquired. We are therefore left to rebuild and validate the full machine learning stack from scratch, a process that is usually tiresome as well as costly. Hence, the re-training approach may not always be the most efficient nor the most effective way for tackling this challenge. Another option is adding patches in the form of wrappers to already deployed models to endow them with new traits or functionalities that help them adapt to the new data conditions, either globally [13] or locally [14]. Another example is that where a company wants to focus on a new client portfolio. This may require evolving from a binary classification setting to a multi-class configuration [15]. In all these cases, structural changes to a model’s environment introduce new operational constraints that cannot be met by the existing solution or a modified version. Instead, it might be necessary to move to a new hypothesis space.

Here, we are concerned with such situations where a drastic change in the demands of a machine learning environment requires some form of adaptation. In this article we study and formalize this problem. We review different strategies that solve the problem of adaptation and categorize them according to the level of access to the data and the degree of knowledge on the source solution internals. Finally, we briefly describe some practical but relevant examples of real situations where the typology of the problem described arises and consider solutions under the umbrella of differential replication.

## 2. Modelling Adaptation to New Environments

The most well known research branch for model adaptation is transfer learning. Transfer learning refers to scenarios where the knowledge acquired when solving one task is recycled to solve a different, yet related task [7]. In general, the problem of transfer learning can be mathematically framed as follows. Given source and target domains $D_{s}$ and $D_{t}$ and their corresponding tasks $T_{s}$ and $T_{t}$, such that $D_{s}\neq D_{t}$, the goal of transfer learning is to build a target conditional distribution $P(y_{t}|x_{t})$ in $D_{t}$ for task $T_{t}$ from the information obtained when learning $T_{s}$ in $D_{s}$. In general, the difference between $D_{s}$ and $D_{t}$ is given by a change in the data distribution, either in the marginal distribution of *x* and *y* or in the joint distribution of both. Observe that the change in any of those distributions directly affects the objective function of the optimization problem. This results in a change in the optimization landscape for the target problem. A graphical illustration of this process is shown in Figure 1a, where the gray and red lines correspond to the source and target optimization objective level sets, respectively. The shaded red area encloses the set of possible solutions for the defined hypothesis space. Transferring the knowledge from source to target requires moving from the original optimum in the source domain to a new optimum in the target domain. This process is done by exploiting the knowledge of the original solution, i.e., by transferring the knowledge between both domains. Advantages of this kind of learning when compared with the traditional scheme are that learning is performed much faster, requiring less data, and even achieving better accuracy results. Examples of methods addressing these issues are pre-training methods [16,17] and warm-start conditioning methods [18,19].

In transfer learning, it usually holds that $T_{s}\neq T_{t}$. There are cases, however, where the task remains the same for both the source and the target domains. This is the case of domain adaptation [10,11]. Domain adaptation is a sub-field of transfer learning that studies cases where there is a change in the data distribution from the source to the target domain. In particular, it deals with learning knowledge representations for one domain such that they can be transferred to another related target domain. This changes of domain can be found, for example, in many of the scenarios resulting from the COVID-19 pandemic. In order to minimize interactions with the points of sale, several countries have decided to extend the limit of transactions where card payments are accepted without requiring cardholders to introduce their pin-code from 20 to 50 euros. Domain adaptation can be of use here to adapt card fraud detection algorithms to the new scenario. Another related branch dealing with adaptation is that of concept drift [20]. In concept drift it is the statistical properties of the target variable that change over time. In general, this happens in the presence of data streams [21]. Under these circumstances adaptive techniques are usually used to detect the drift and adjust the model to the new incoming data from the stream.

Here, we focus in an altogether different adaptation problem. In our described scenario, the task remains the same, $T_{s}=T_{t}$, but changes in the environmental conditions renders current solution non apt for the task. The new environmental conditions can be formally defined as a set of new constraints, C, added to the problem. As a result of these constraints the solution in the source scenario lays outside of the feasible set. The adaptation problem consists of finding a new solution that satisfies these constraints.

We can frame this problem using the former notation as follows. Given a domain D, its corresponding task T, and the set of original environmental constraints $C_{s}$ that make the solution of this problem feasible, we assume an scenario were a hypothesis space $H_{s}$ has already been defined. In this context, we want to learn a new solution for the same task and domain, but for a new target scenario defined by a new set of feasibility constraints $C_{t}$, where $C_{t}\neq C_{s}$. In the most general case, solving this problem requires the definition of a new hypothesis space $H_{t}$. In a concise form and considering an optimization framework this can be rewritten as follows, where Scenario I corresponds to the source domain and constraints and Scenario II describes the new conditions
Scenario I
forTinDmaximizeforh∈Hs P(y|x;h)subject to Cs→Scenario IIforTinDmaximizeforh∈Ht P(y|x;h)subject to Ct

We identify the problem above as that of environmental adaptation. Under the above notation, the initial solution, the existing optimum, corresponds to a model $h_{s}$ that belongs to the hypothesis space $H_{s}$ defined for the first scenario. This is a model that fulfills the constraints $C_{s}$ and maximizes P(y|x;h) for a training dataset S={(x,y)}, defined by task T on the domain D. Adaptation involves transitioning from this scenario to *Scenario II*, a process which may be straightforward, although this is not always the case.

Take, for example, the two cases displayed in Figure 1b,c. In this figure, the optimization objective level sets defined by the domain and the task is displayed in gray, while the shaded area corresponds to the defined hypothesis space $H_{s}$. The rectangles shown in red refer to the new feasible set defined by the constraints imposed by the environment, $C_{t}$. Observe that in both figures, the source solution (in gray) is not feasible for the target scenario. In Figure 1b the new feasible set defined is compatible with the existing hypothesis space. Hence, environmental adaptation may simply imply finding a new optimum in this space that comply with the constraints. In contrast, there are cases where the whole set of solutions defined by the source hypothesis space is unfeasible in the target scenario. This happens when there is no overlap between the feasible set defined by target constraints $C_{t}$ and the set of models defined by the source hypothesis space, $H_{s}$. In such cases, adaptation requires that we define an altogether new hypothesis space $H_{t}$. An example of this is shown in Figure 1c, where the constraints exclude the models in $H_{s}$ from the set of possible solutions. Hence, we need to define a new hypothesis space that is compatible with the new environment and where we can find an optimal solution for the given domain and task.

We stress that this problem is different to that of transfer learning and domain adaptation. For both these settings, the solution in the source domain, while sub-optimal, is generally still feasible in the target domain. In environmental adaptation, however, the solution in the source scenario is unfeasible in the target scenario. For illustration purposes consider the case of a multivariate Gaussian kernel support vector machine. Assume that due to changes in the existing regulation, this model is required to be fully interpretable in the considered application. The new set of constraints is not compatible with the source scenario and hence we would require a complete change of substrate, i.e., a new hypothesis space. Section 4 develops this case and introduces more examples of the need of environmental adaptation.

For cases such as these, we introduce the notion of differential replication of machine learning models as an efficient approach to ensuring environmental adaptation. In particular, differential replication enables model survival in highly demanding environments, by building on the knowledge acquired by previously trained models in generations. This effectively involves solving the optimization problem for Scenario II considering the solution obtained for Scenario I.

## 3. Differential Replication

“When copies are made with variation, and some variations are in some tiny way “better” (just better enough so that more copies of them get made in the next batch), this will lead inexorably to the ratcheting process of design improvement Darwin called evolution by natural selection.” [Daniel Dennett, Breaking the Spell, p. 78, 2006]

Sometimes, solving the environmental adaptation problem can be straightforwardly achieved by discarding the existing model and re-training a new one. However, it is worth considering the costs of this approach. In general, rebuilding a model from scratch (i) implies obtaining the clearance from the legal, business, ethical, and engineering departments, (ii) does not guarantee that a good or better solution of the objective function will be achieved (The objective function in this scenario corresponds to P(y|x;h)), (iii) requires a whole new iteration of the machine learning pipeline, which is costly and time-consuming, (iv) assumes full access to the training dataset, which may no longer be available or require a very complex version control process. Many companies circumvent these issues by keeping machine learning solutions up-to-date using automated systems that continuously evaluate and retrain models, a technique known as continuous learning. Note, however, that this may take huge storage space, due to the need to save all the new incoming information. Hence, in the best case scenario, re-training is an expensive and difficult approach that assumes a certain level of knowledge that is not always guaranteed. In what follows we consider other techniques.

Under the theory of Natural Selection, environmental adaptation relies on changes in the phenotype of a species over several generations to guarantee its survival in time. This is sometimes referred to as differential reproduction. In the same lines, we define differential replication of a machine learning model as a cloning process in which traits are inherited from generation to generation of models, while at the same time adding variations that make descendants more suitable for the new environment. More formally, differential replication refers to the process of finding a solution $h_{t}$ that fulfills the constraints $C_{t}$, i.e., it is a feasible solution, while preserving/inheriting features from $h_{s}$. In general, $P\left(y|x;h_{t}\right)∼P\left(y|x;h_{s}\right)$, so that in the best case scenario, we would like to preserve or improve the performance of the source solution $h_{s}$, here referred to as the parent. However, this is a requirement that may not always be achieved. In a biological simile, requiring a cheetah to be able to fly may imply losing its ability to run fast. In what follows we consider existing approaches to implement differential replication in its attempt to solve the problem of environmental adaptation.

### Differential Replication Mechanisms

The notion of differential replication is built on top of two concepts. First, there is some inheritance mechanism that is able to transfer key aspects from the previous generation to the next. That would account for the name of replication. Second, the next generation should display new features or traits not present in their parents. This corresponds to the idea of differential. These new traits should make the new generation more fit to the environment to enable environmental adaptation of the offspring.

Particularizing to machine learning models, implementing the concept of differential may involve a fundamental change in the substratum of the given model. This means we might need to define a new hypothesis space that fulfills the constraints of the new environment $C_{t}$. Consider, for example, the case of a large ensemble of classifiers. In highly time demanding tasks, this model may be too slow to provide real time predictions when deployed into production. Differential replication enables moving from this architecture to a simpler, more efficient one, such as that of a shallow neural network [22]. This "child" network can inherit the decision behavior of its predecessor while at the same time being better adapted to the new environment. Conversely, replication requires that some behavioral aspect be inherited by the next generation. Usually, it is the model’s decision behavior that is inherited, so that the next generation will replicate the parent decision boundary. Replication can be attained in many different ways. As shown in Figure 2, depending on the amount of knowledge that is assumed about the initial data and model, mechanisms for inheritance can be grouped under different categories.

Inheritance by sharing the dataset:Two models trained on the same data are bound to learn similar decision boundaries. This is the weakest form of inheritance possible, were no actual information is transferred from source to target solution. Here the decision boundary is reproduced indirectly and mediated through the data themselves. Re-training falls under this category [23]. This form of inheritance requires no access to the parent model, but assumes knowledge of its training data. In addition to re-training, model *wrappers* can be used to envelope the existing solutions with an additional learnable layer that enables adaptation [13,24] (It is worth mentioning that the family of wrappers may require access to the model internals. In this study we classify them in this category by considering the most agnostic and general case).Inheritance using edited data: Editing is the methodology that allows data selection for training purposes [25,26,27]. Editing can be used to preserve those data that are relevant to the decision boundary learned by the original solution and use them to train the next generation. Take, for example, the case where the source hypothesis space corresponds to the family of support vector machines. In training a differential replica, one could retain only those data points that were identified as support vectors [28]. This mechanism assumes full access to the model internals, as well as to the training data.Inheritance using model driven enriched data: Data enrichment is a form of adding new information to the training dataset through either the features or the labels. In this scenario, each data sample in the original training set is augmented using information from the parent decision behavior. For example, a sample can be enriched by adding additional features using the prediction results of a set of classifiers. Alternatively, if instead of learning hard targets one considers using the output of the parent’s class probability outputs or logits as soft-targets, this richer information can be exploited to build a new generation that is closer in behavior to the parent. Under this category fall methods like model distillation [22,29,30,31], as well as techniques such as label regularization [32,33] and label refinery [34]. In general, this form of inheritance requires access to the source model and is performed under the assumption of full knowledge of the training data.Inheritance by enriched data synthesis: A similar scenario is that where the original training data are not accessible, but the model internals are open for inspection. In this situation, the use of synthetic datasets has been explored [22,35]. In some cases, intermediate information about the representations learned by the source model are also used as a training set for the next generation. This form of inheritance can be understood as a zero-shot distillation [36].Inheritance of model’s internal knowledge: In some cases, it is possible to access the internal representations of the parent model, so that more explicit knowledge can be used to build the next generation [37,38]. For example, if both parent and child are neural networks, one can force the mid-layer representations to be shared among them [39]. Alternatively, one could use the second level rules of a decision tree to guide the next generation of rule-based decision models.Inheritance by copying: In highly regulated environments, access to the original training samples or to the model internals may not be possible. In this context, experience can also be transmitted from one model to its differential replica using synthetic data points labelled according to the hard predictions of the source model. This has been referred to as copying [40,41].

Note that on top of a certain level of knowledge about either the data or the model, or both, some of the techniques listed above often impose also additional restrictions on the considered scenarios. Techniques such as distillation, for example, assume that the original model can be controlled by the data practitioner, i.e., internals of the model can be tuned to force specific representations of the given input throughout the adaptation process. In certain environments this may be possible, but generally it is not.

## 4. Differential Replication in Practice

In what follows we describe seven different scenarios where differential replication can be exploited to ensure a devised machine learning solution adapts to changes in its environment. In all seven of them we assume an initial model has already been trained and served into production. This model and its characteristics correspond to Scenario I, as defined above. We describe how the constraints that apply to Scenario II differ from the original scenario and discuss different techniques and approaches to adapting the existing solution to the new requirements. Note that, while specific specific examples are given here, other solutions based on differential replication may also be possible.

### 4.1. Moving to a Different Software Environment

Model deployment is often costly in company environments [42,43,44,45]. Common issues include the inability to maintain the technological infrastructure up-to-date with latest software releases, conflicting versions or incompatible research and deployment environments. Indeed, in-company infrastructure is subject to continuous updates due to the rapid pace with which new software versions are released to the market. At any given time, changes in the organizational structure of a company may drive the engineering department to change course. Say, for example, that a company whose products were originally based on Google’s Tensorflow package [46] makes the strategic decision of moving to Pytorch [47]. In doing so, they might decide to re-train all models from scratch in the new environment. This is a long and costly process that can result in a potential loss of performance. Specially if the original data are not available or the in-house data scientists are new to this framework. Alternatively, using differential replication the knowledge acquired by the existing solutions could be exploited in the form of hard or soft labels or as additional data attributes for the new generation.

Equivalently, consider the opposite case, where a company previously relying on other software now decides to train its neural network models using Tensorflow. Despite the library itself provides detailed instructions on how to serve models in production [48], this typically requires several third-party components for docker orchestration, such as Kubernetes or Elastic Container Service [49], which are seldom compatible with on-premise software infrastructure. Instead, exploiting the knowledge acquired by the neural network to train a child model in a less demanding environment may help bridge the gap between the data science and engineering departments.

### 4.2. Adding Uncertainty to Prediction Outputs

In applications where machine learning models are used to aid in high-stakes decisions, producing accurate predictions may not always be enough. In those applications information about the risks or confidence associated with predictions may be required. This is the case, for example, of medical diagnosis [50]. Consider a case where an existing machine learning solution produces only hard predictions. In this situation, doctors and data practitioners have very little information on what the level of confidence is behind each output. Yet, a new protocol may require refraining from making predictions in cases of large uncertainty. To meet this new requirement, a new learnable algorithmic component can be added to wrap the original solution and endow it with a layer of uncertainty to measure the confidence in prediction [13,24,51].

### 4.3. Mitigating the Bias Learned by Trained Classifiers

Machine learning models tend to reproduce existing patterns of discrimination [52,53]. Some algorithms have been reported to be biased against people with protected characteristics like ethnicity [54,55,56,57], gender [58,59] or sexual orientation [60]. As a model is tested against new data throughout its lifespan, some of its learned biases may be made apparent [61]. Consider one of such scenarios, where a deployed model is found to be biased in terms of a sensitive attribute. Under such circumstances, one may wish to transit to a new model that inherits the original predictive performance but which ensures non-discriminatory outputs. A possible option is to edit the sensitive attributes to remove any bias, therefore reducing the disparate impact in the task T, and then training a new model on the edited dataset [62,63]. Alternatively, in very specific scenarios where the sensitive information is not leaked through additional features, it is possible to build a copy by removing the protected data variables [64]. Or even, to redesign the hypothesis space considering a loss function that accounts for the fairness dimension when training subsequent generations.

### 4.4. Evolving from Batch to Online Learning

In this scenario we consider the transition from a batch classifier to an adaptive classifier capable of handling concept drift. In general, companies train and deploy batch learning models. However, these are very rapidly rendered obsolete by their inability to adapt to a change in the data distribution. When this happens, the most straightforward solution is to wait until there are enough samples of the new distribution and re-train the model. However, this solution is timely and often expensive. A faster solution to ensure adaptation to the new data distribution is to use the idea of differential replication to create a new enriched dataset able to detect the data drift. For example, including the soft targets and a timestamp attribute in the target domain, $D_{t}$. One may then use this enriched dataset to train a new model that replicates the decision behavior of the existing classifier. To allow this new model to also learn from new incoming data samples we may additionally incorporate the online requirement in the constraints $C_{t}$ for the differential replication process [65].

### 4.5. Preserving the Privacy of Deployed Models

Developing good machine learning models requires abundant data. The more accessible the data, the more effective a model will be. In real applications, training machine learning models requires collecting a large volume of data from users, often including sensitive information. When models trained on user data are released and made accessible through specific APIs, there is a risk of leaking sensitive information. Differential replication can be used to avoid this issue by training another model, usually a simpler one, that replicates the learned decision behavior but which preserves the privacy of the original training samples by not being directly linked to these data. The use of distillation techniques in the context of teacher-student networks, for example, has been reported to be successful in this task [66,67].

In order to minimize the risk of leaking personal data through models, the European General Data Protection Regulation [68] recognizes the principle of data minimization, which dictates that personal data shall be limited to what is necessary in relation to the purposes for which they are processed. However, it is often difficult to determine the minimum amount of data required. Differential replication has been shown to be successful in this task by producing a generalization of the model that reduces the amount of personal data needed to obtain accurate predictions [69].

### 4.6. Intelligible Explanations of Non-Linear Phenomena

Recent advances in the field of machine learning have led to increasingly sophisticated models, capable of learning ever more complex problems to a high degree of accuracy. This comes at the cost of simplicity [70,71], a situation that stands in contrast to the growing demand for transparency in automated processing [68,72,73]. As a result, a widely established technique to provide explanations is to use linear models, such as logistic regression. Model parameters, i.e., the linear coefficients associated to the different attributes, can then be used to provide explanations to different audiences. Although this approach works in simple scenarios where the variables do not need to be modified nor pre-processed, this is seldom the case for real life applications, where variables are usually redesigned before training and new more complex features are introduced. This is even worse when, in order to improve model performance, data scientists create a large set of new variables, such as bi-variate ratios or logarithm scaled variables, to capture non-linear relations between original attributes that linear models cannot handle during the training phase. This results in new variables being obfuscated and therefore often not intelligible for humans.

Recent papers have shown that the knowledge acquired by black-box solutions can be transferred to interpretable models such as trees [74,75,76], rules [77] and decision sets [78]. Hence, a possible solution to the problem above is to replace the whole predictive system, composed by both the pre-processing/feature engineering step and the machine learning model by a copy that considers both steps as a single black box model [79]. Doing this, we are able to deobfuscate model variables by training copies to learn the decision outputs of trained models directly from the raw data attributes without any pre-processing. Another possible approach is using wrappers. This is, for example, the case of LIME [14], where a local interpretable proxy model is learned by perturbing the input in the neighborhood of a prediction and using the original solution as a query oracle.

### 4.7. Model Standardization for Auditing Purposes

Auditing machine learning models is not an easy task. When an auditor wants to audit several models under the same constraints all audited models need to fulfill an equivalent set of requirements. Those requirements may limit the use of certain software libraries, or of certain model architectures. Usually, even within the same company, each model is designed and trained on its own basis. As research in machine learning grows, new models are continuously devised. However, this fast growth in available techniques hinders the possibility of having a deep understanding of the mechanisms underlying the different options and makes the assessment of some auditing dimensions a nearly impossible task.

In this scenario, differential replication can be used to establish a small set of canonical models into which all others can be translated. In this sense, a deep knowledge of these set of canonical models would be enough to conduct auditing tests. Say, for example, that we define the canonical model to be a deep learning architecture with a certain configuration. Any other model can be translated into this particular architecture using differential replication (provided the capacity of the network is large enough to replicate the given decision boundary). The auditing process need then only consider how to probe the canonical deep network to report impact assessment.

## 5. Conclusions

In this paper, we have formalized the problem of environmental adaptation, which refers to situations where changes in the environment surrounding a model make current solution unusable and require it to adapt to new constraints. To tackle this issue we define a mechanism inspired in how biological organisms evolve: differential replication. Differential replication allows machine learning models to modify their behavior to meet the new requirements defined by the environment. We envision this replication mechanism as a projection operator able to translate the decision behavior of a machine learning model into a new hypothesis space with different characteristics. Under this term we group different techniques previously described in the literature, here referred to as inheritance mechanisms. We provide a categorization of these mechanisms in terms of the considered setting. These range from the more permissive inheritance by sharing the dataset to the more restrictive inheritance by copying, which is the solution requiring less knowledge about the parent model and training data. Finally, we provide several examples of how differential replication can be applied in practice to solve the environmental adaptation problem in seven different real-life scenarios.

## Figures and Tables

**Figure 1 entropy-22-01122-f001:**
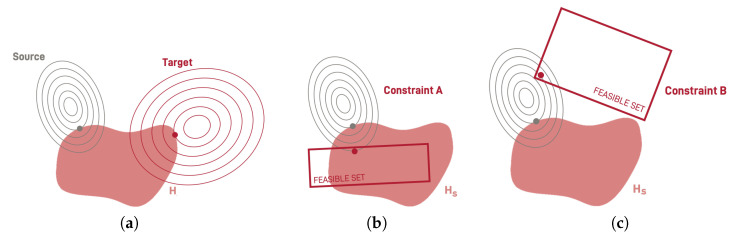
The problems of (**a**) transfer learning and environmental adaptation for (**b**) a case where the new new feasible set overlaps with part of the existing hypothesis space and (**c**) a case where there is no such overlap. The gray and red lines and dots correspond to the set of possible solutions and the obtained optimum for the source and target domains, respectively. The shaded area shows the defined hypothesis space.

**Figure 2 entropy-22-01122-f002:**
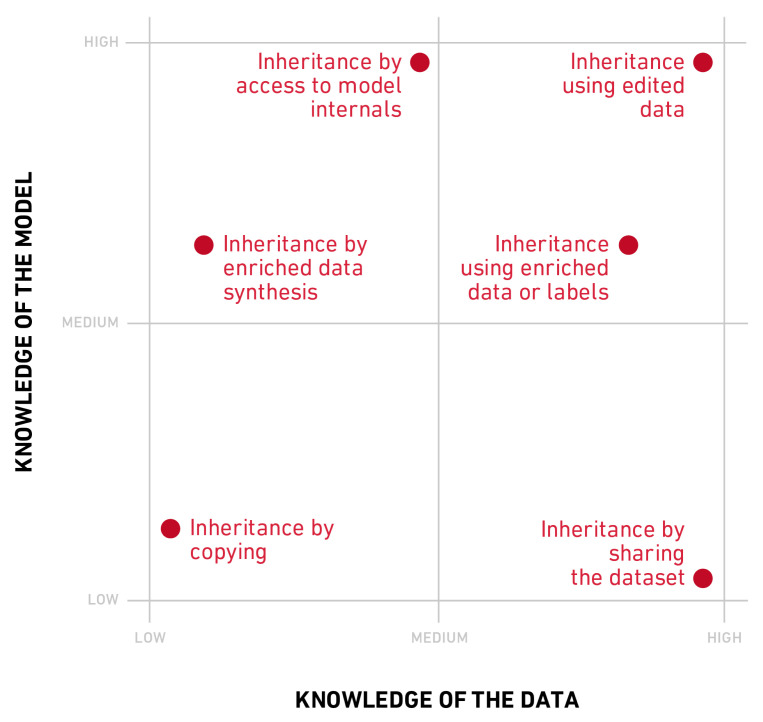
Inheritance mechanisms in terms of their knowledge of the data and the model internals.

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
