# Peer review of "Environmental Adaptation and Differential Replication in Machine Learning"

_entropy, 2020, doi:10.3390/e22101122_

Round 1
Reviewer 1 Report
This paper discusses a general framework referred to as "differential replication" to evolve new generations of machine learning models, when the operating environment or the requirements change after model deployment. The authors attempt to organize some existing "knowledge reuse" mechanisms under this new framework and some practical considerations (e.g. interpretability, privacy etc.) are discussed.
Though the paper is reasonably easy to follow, it has a number of fundamental issues that which I discuss below in detail.
- Is this paper positioned as an original research article or a survey paper? If it is the former the paper is very low on novelty. On the other hand, if it is the latter, the exposition is not broad enough to cover the relevant methods in the context of knowledge reuse. Consequently, it is not publishable in its current form.
- Lack of clarity in its objectives: Is the primary objective to setup a general taxonomy to organize the different knowledge transfer methods? Why is that needed for the ML community? Why is the existing classification of methods insufficient?
- What does the term environmental constraints mean? How is the term ecosystem defined? In the context of this paper, what does interpretability entail?
- The problems setup is not clear. Do the authors access to labeled data in the target environment/domain/task? Is the challenge in obtaining labeled data in the target domain?
- In the main formulation (line 95), do the (x,y) refer to the same data? Since the same term T is used, is the task also the same across the two scenarios?
- The only example provided for the environmental change is need for interpretability and in the later part of the paper the authors allude to building a linear model as the next generation model. How is this different from existing explainable AI methods including LIME, MUSE, ROPE, CXPlain etc.
- I strongly urge the authors to avoid non-rigorous statements to make the paper suitable for research audience. For example,
- (line 37) over 60% of machine learning models developed in companies are never actually put in production - What kind of models are being referred to here and why are they not deployed?
- (line 55) In contrast, when the nature of the system that changes, alternatives are scarce - what nature is being referred to here?
Author Response
Please find attached our responses to your comments

Reviewer 2 Report
This paper proposes a solution based on reusing the knowledge acquired by the already deployed machine learning models and leveraging it to train future generations.
Just a divulgative paper.
Lacks of some algorithm, some data to test, testing labs, etc...
Interesting ideas but no fact they work.
Author Response
Please find attached our response to your comments

Round 2
Reviewer 2 Report
All reviewer's comments have been addressed. The paper has been improved according made changes.
Just some minor changes before proof editing.
Lines 3,5,7,9, etc. and along the paper --> Use impersonal form "we.." --> "this paper ..", "this work ...", "Proposed model ...", etc.
Line 23, 290, 304, 315, etc., and also in some references. --> "...for survival[1]. Survival..." --> include a blank/space before reference "...for survival [1]..."
Figure 2.- Use black color in axis labels (low, medium, high)
Line 422. An strange "." appears as the first char in paragraph